# Research on the Evolution of the Economic Spatial Pattern of Urban Agglomeration and Its Influencing Factors, Evidence from the Chengdu-Chongqing Urban Agglomeration of China

Rui Ding [1,2,†] , Jun Fu [1,2,†], Yiling Zhang [1,2,*], Ting Zhang [1,2], Jian Yin [3], Yiming Du [1,2], Tao Zhou [1,2] and Linyu Du [1,2]

1   College of Big Data Application and Economics (Guiyang College of Big Data Finance), Guizhou University of Finance and Economics, Guiyang 550025, China
2   Key Laboratory of Green Fintech, Guizhou University of Finance and Economics, Guiyang 550025, China
3   School of Water Conservancy and Civil Engineering, Northeast Agricultural University, Xiangfang District, Harbin 150050, China
*   Correspondence: evalinzyl@mail.gufe.edu.cn
†   Co-first author.

**Abstract:** To investigate the spatial evolution process of economic development in the urban agglomeration and its influencing factors, the network construction method, modified gravity model, geographic detector and Geographically Weighted Regression (GWR) model are used to analyze the intensity of urban association; then, the evolution of economic, spatial pattern and its influencing factors are further discussed, and the Chengdu-Chongqing urban agglomeration of China from 2005 to 2020 is studied as an example. The results show that: (1) the economically developed zones of the Chengdu-Chongqing urban agglomeration mainly concentrated in the core cities of Chengdu and the central city of Chongqing, and the region shows an uneven spatial pattern of economic development distribution. (2) The share of economic linkages with the central city of Chengdu and Chongqing as the twin cities is significant, the intensity of Chengdu and its neighboring cities is gradually decreasing, while the central city of Chongqing has increased, but it still has an insufficient influence on the peripheral areas. (3) The intensities and directions of the factors influencing economic development in the Chengdu-Chongqing urban agglomeration are different. The total output value of the secondary industry, total social fixed asset investment, the number of beds in health institutions, and road freight turnover are significant factors with consistently strong explanatory ability for economic development. The promotion effect of these four significant factors on economic development is mainly concentrated in the eastern and western regions of Chengdu-Chongqing urban agglomeration, while the inhibiting effect is mainly on the cities in the south and north. Based on this study, relevant recommendations are made to promote the coordinated development of the Chengdu-Chongqing urban agglomeration.

**Keywords:** economic spatial pattern; gravity model; geographic detector; geographically weighted regression; Chengdu-Chongqing urban agglomeration

## 1. Introduction

As an advanced phenomenon of regional economic and spatial forms in the process of industrialization, urban agglomeration is a landmark product of a certain stage of urban development, and urban agglomeration planning has become an important strategic development deployment in China. In 2016, China issued the Chengdu-Chongqing Urban Agglomeration Development Plan, which states that the Chengdu-Chongqing urban agglomeration is an important platform for the development of the western region, strategic support for the Yangtze River Economic Belt, and an important demonstration area for China to promote a new type of urbanization. Due to the huge differences in

education and health levels, trade and commerce service capacity, transport infrastructure construction and investment environment among the cities, there are enormous dissimilarities in the economic development levels of the cities in the urban agglomeration. Therefore, it is of great significance to scientifically grasp the economic development level of each city in the Chengdu-Chongqing urban agglomeration and analyze its influencing factors to promote the coordinated development of the urban agglomeration and optimize its spatial layout.

As for the spatial pattern of regional economies, the existing research is mainly carried out in the following two directions. The first direction is the structuralist narration based on the calculation of spatial differences and the description of the current situation of spatial layout. Some literature uses indicators such as urban resident population, per capita GDP, total retail sales of social consumer goods, the added value of the primary industry, added value of secondary industry and added value of the tertiary industry. ESDA [1–3], Gini coefficient [4–7], coefficient of variation [8], entropy method [9], Theil index [8], gravity model [10,11], social network analysis [12] and other methods are used to carry out the research. For example, Dong et al. [2] used the ESDA method to analyze the spatial pattern of county economies in the Chang-Zhu-Tan urban agglomeration of China, answering whether the county economies in the Chang-Zhu-Tan urban agglomeration developed according to the integration policy. Chen et al. [8] analyzed the spatial and temporal characteristics of the urban–rural income gap and its driving forces in the Yangtze River Economic Belt from 2000–2017 using the coefficient of variation and the Theil index and found that the spatial divergence pattern of the urban–rural income gap is influenced by both natural and socio-economic factors; of these, socio-economic factors predominate. The second direction is the relational narrative based on cities' connection and network construction. At present, there are three different perspectives: economic connection, transportation network, and information flow. For example, Ye et al. [13] used the urban flow intensity to modify the traditional gravity model to construct the economic connection matrix to analyze the evolution characteristics of the economic connection network structure of Guanzhong Plain urban agglomeration and its impact on economic growth. Guo et al. [14] used the complex network analysis method to further analyze the characteristics of urban connection networks in Northeast China from the changes of the "high-speed railway+" network in different periods, through the comparison between "high-speed railway+" networks and high-speed railway networks. Qiao et al. [15] analyzed the characteristics and influencing factors of an urban spatial connection and network structure of urban agglomeration in the Yangtze River Delta from the perspective of information flow. At the scale of urban agglomeration, the research on inter-city network structure based on economic spatial pattern and economic connection mainly focuses on Beijing-Tianjin-Hebei [16–18] and the Yangtze River Delta regions [16,19–21]. Some scholars have discussed the factors affecting economic development from different perspectives, such as Zhao et al. [22] from the perspective of the energy-economy, focused on the estimation of energy economic efficiency of the Yangtze River Urban Agglomeration (YRUA) and decomposed the energy-economic efficiency into pure technical efficiency and scale efficiency to better explore the restrictive factors for the improvement of energy-economic efficiency. Wei et al. [23] studied the evolution of marine industrial structures and analyzed their impact on the maritime economy's Green Total Factor Productivity (GTFP). Chen et al. [24] explored the drivers of the tourism economic network structure formation on the Qinghai-Tibet Plateau from 2015–2019, using the gravity model and social network analysis, and they have found that A-class attractions and star-rated hotels significantly contributed to spatial associations. From the spatial dimension, they used several indicators to comprehensively evaluate the development of regional tourism employment through horizontal and vertical comparison; their results show that the regional tourism economic growth is driven by investment.

The above-mentioned studies on the spatial pattern of the regional economy have all achieved fruitful results, but there is still some room for expansion. Firstly, there are more studies on the Beijing-Tianjin-Hebei and Yangtze River Delta regions, but there is a

lack of studies on the Chengdu-Chongqing urban agglomeration. Secondly, most studies have mainly explored the evolution of economic spatial patterns on a spatial scale, with relatively few studies combining temporal and spatial scales, and there is a lack of analysis of the direction and effects of the drivers affecting the level of economic development. Based on this, this paper uses the modified gravity model to take the 36 cities/counties of the Chengdu-Chongqing urban agglomeration as the research object, and selects four time periods, which are 2005, 2010, 2015, and 2020. Combined with the time and spatial scale, this paper explores the evolution characteristics of the economic spatial pattern of the Chengdu-Chongqing urban agglomeration. It uses the method of geographic detector combined with the GWR model to analyze the impact direction and effect of its driving factors on economic development, to provide some suggestions for the collaborative economic development of Chengdu-Chongqing urban agglomeration.

## 2. Data and Methods

### 2.1. Overview of the Study Area

This paper takes the Chengdu-Chongqing urban agglomeration as the research object. It adopts the scope stipulated in the Chengdu-Chongqing Urban Agglomeration Development Plan jointly issued by the National Development and Reform Commission and the Ministry of Housing and Construction in 2016, with 15 prefecture-level cities in Sichuan and 21 districts and counties in Chongqing as the research objects (Figure 1). As the names of administrative units in Chongqing differ slightly between 2005 and 2020, the latest administrative unit names from the 2021 Chongqing Statistical Yearbook are used in this paper. The central urban area of Chongqing includes Yuzhong District, Dadukou District, Jiangbei District, Shapingba District, Jiulongpo District, Nan'an District, Beibei District, Yubei District, and Banan District, which are combined into one zone. Chengdu-Chongqing urban agglomeration is located in southwest China, with a total area of about 185,000 km², with a resident population of 100.7 million in 2020, accounting for 6.9% of the country. The regional GDP is 6.8 trillion, accounting for 6.7% of the country. It is one of the regions with the best economic foundations and the strongest economic strength intensity in the west. It has the regional advantage of connecting the east and the west and connecting the north and the south. At the same time, it has an excellent endowment of natural resources, strong comprehensive carrying capacity, strong foundation of manufacturing business, finance, and other industries, high degree of openness, rich human resources, and a good innovation and entrepreneurship environment. It is a typical region with national economic importance and strong network connection characteristics.

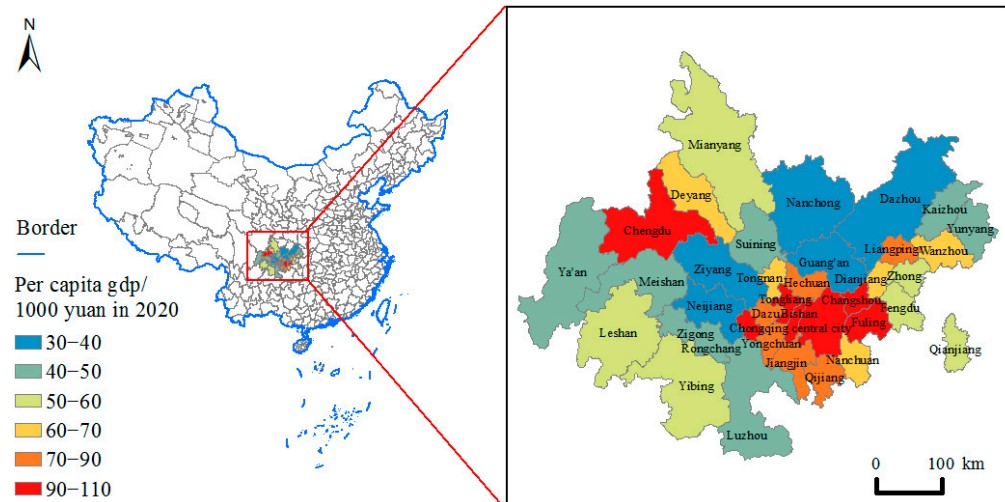

**Figure 1.** Research area and GDP per capita distribution.

### 2.2. Selection of Indicators and Data Sources

This paper selects four time periods from 2005 to 2020 to analyze the evolution of the economic spatial pattern of the Chengdu-Chongqing urban agglomeration, explore the action intensity of factors affecting economic development and analyze the spatial heterogeneity of significant factors. The urban comprehensive central system is divided into four subsystems: production and employment, business services, education and health, and transportation. Due to the availability and accuracy of the data, the production and employment center selects urban non-private sector employees (10,000), rural employees (10,000), the total output value of primary industry (100 million yuan) and the total output value of secondary industry (100 million yuan) for calculation. The business services center selects total social fixed asset investment (100 million yuan), total retail sales of social consumer goods (100 million yuan) and the total output value of tertiary industry (100 million yuan) are selected for calculation. The number of ordinary primary and secondary schools, number of pupils in general primary and secondary schools, the number of health institutions and the number of beds in health institutions are selected by the education and health center for calculation. Total road mileage (km), road passenger turnover (10,000 km) and road freight turnover (million ton kilometers) are selected for traffic center calculation. Through these 14 evaluation indexes, the urban centrality of the Chengdu-Chongqing urban agglomeration is comprehensively evaluated.

The centrality of the four subsystems is obtained by calculating these 14 comprehensive evaluation indicators, and the urban comprehensive centrality score is obtained by summing them up. Then the urban economic correlation intensity is further calculated based on the urban comprehensive centrality score and the geographical distance between cities. The indicators in four years are from Sichuan statistical yearbook, Chongqing statistical yearbook, and Chongqing traffic Yearbook; the missing values of some index data shall be replaced by similar years.

### 2.3. Research Methodology

The comprehensive centrality of the city and the improved gravity model is used to calculate the strength of urban economic correlation. At the same time, combined with the geographic detector and GWR model, this paper explores the significant driving factors affecting the economic development level of the Chengdu-Chongqing urban agglomeration.

2.3.1. Network Construction Methods

Step 1, to establish the initial matrix. The research object is the 36 cities/districts and counties in the Chengdu-Chongqing urban agglomeration; each research object uniformly selects 14 indicators, the initial matrix can be set as $x = \{x_{ij}\}36 \times 14$ ($1 \leq i \leq 36$, $1 \leq j \leq 14$), $x_{ij}$ represents the $j$-th indicator of the $i$-th city.

Step 2, to normalize the data. The standard matrix can be set as $y = \{y_{ij}\}36 \times 14$, with an ideal value set as $x'_{ij}$ and $y_{ij}$ calculated as:

$$x_{ij} = \frac{x - x_{min}}{x_{max} - x_{min}}, \tag{1}$$

$$y_{ij} = \frac{x'_{ij}}{\sum x'_{ij}}. \tag{2}$$

Step 3, to calculate the entropy value $e_j$, the variability index $g_i$ and the weight $w_j$ for each indicator, calculated by the formula:

$$e_j = -k \sum y_{ij} ln y_{ij}, \tag{3}$$

$$k = \frac{1}{lnm}, \tag{4}$$

$$g_j = 1 - e_j, \tag{5}$$

$$w_j = \frac{g_j}{\sum g_j}. \tag{6}$$

Step 4, to calculate the index $c_{ij}$ of the $j$-th indicator for the $i$-th city based on the weight $w_j$ of each indicator and the standardization matrix $y = \{y_{ij}\}36 \times 14$. The formula is as follows:

$$c_{ij} = w_j \times x'_{ij}. \tag{7}$$

Step 5, to add each index to get the central comprehensive index $c_i$ of each city, which can be expressed as:

$$c_i = \sum c_{ij}. \tag{8}$$

### 2.3.2. Gravitational Model

The study of applying the gravity model to the scope of urban economic influence is called the urban gravity model. The inter-city distance $d_{ij}$ and the combined city centrality score was chosen to calculate the city correlation intensity $R$, which is calculated as:

$$R = p \times \frac{c_i^a \times c_j^b}{d_{ij}^r} \tag{9}$$

where $R$ denotes the economic correlation between two cities, $c_i$ and $c_j$ represent the combined centrality scores of city $i$ and city $j$ respectively; as only one city size indicator of city centrality has been chosen, it can be assumed that $a = b = 1$. This model is mainly used to measure the gravitational force between two cities, here $p$ is constant 1; $r$ is the distance decay coefficient, regarding other related studies, $r$ is taken as 2; $d_{ij}$ represents the distance between city $i$ and city $j$, where $d_{ij}$ can be measured according to the "point distance" tool in ArcGIS.

### 2.3.3. Geo-Detectors

The geographic detector is a spatial statistical method to reveal the spatial differentiation of geographical elements and the driving forces. To explore the factors affecting the economic development of the Chengdu-Chongqing urban agglomeration, the factor_detector is used for analysis to clarify the statistically significant independent variables and their explanatory strength of the dependent variables, measured by $q$-values, calculated by the formula [25]:

$$q = 1 - \frac{\sum_{h=1}^{L} N_h \sigma_h^2}{N \sigma^2} \tag{10}$$

where $h = 1, 2, 3, \ldots, L$ is the stratification of variable Y or factor X, that is, classification or partition; $N_h$, $N$ is the number of cells in stratum h and the whole region respectively; $\sigma_h^2$, $\sigma^2$ are the variances of variable Y in stratum $h$ and the whole area respectively.

### 2.3.4. Geographically Weighted Regression (GWR)

GWR is a spatial regression model based on the idea of local smoothing, which can not only effectively estimate the data with spatial autocorrelation, but also reflect the spatial heterogeneity of parameters in different regions. The model formula is:

$$y(u) = \beta_0(u) + \sum_{k=1}^{p} \beta_k(u) \times x_k(u) + \varepsilon(u) \tag{11}$$

where $\beta_0(u)$ is the intercept term; $\beta_k(u)$ is the regression coefficient of the $k$-th covariate; $x_k(u)$ is the value of the $k$-th covariate at position $u$; $p$ is the number of regression terms; $\varepsilon(u)$ is the random error term at position $u$.

## 3. Analysis of the Spatial Pattern of the Chengdu-Chongqing Urban Agglomeration

In the ArcGIS10.7 software, using the Natural breakpoint method, each centrality is divided into six levels, namely Low-value area, Sub-low-value area, Lower-value area, Higher-value area, Sub-high-value area, High-value area. Through map visualization, it

can clearly show the dominant areas and the differences between regions. As shown in Figure 2a–t.

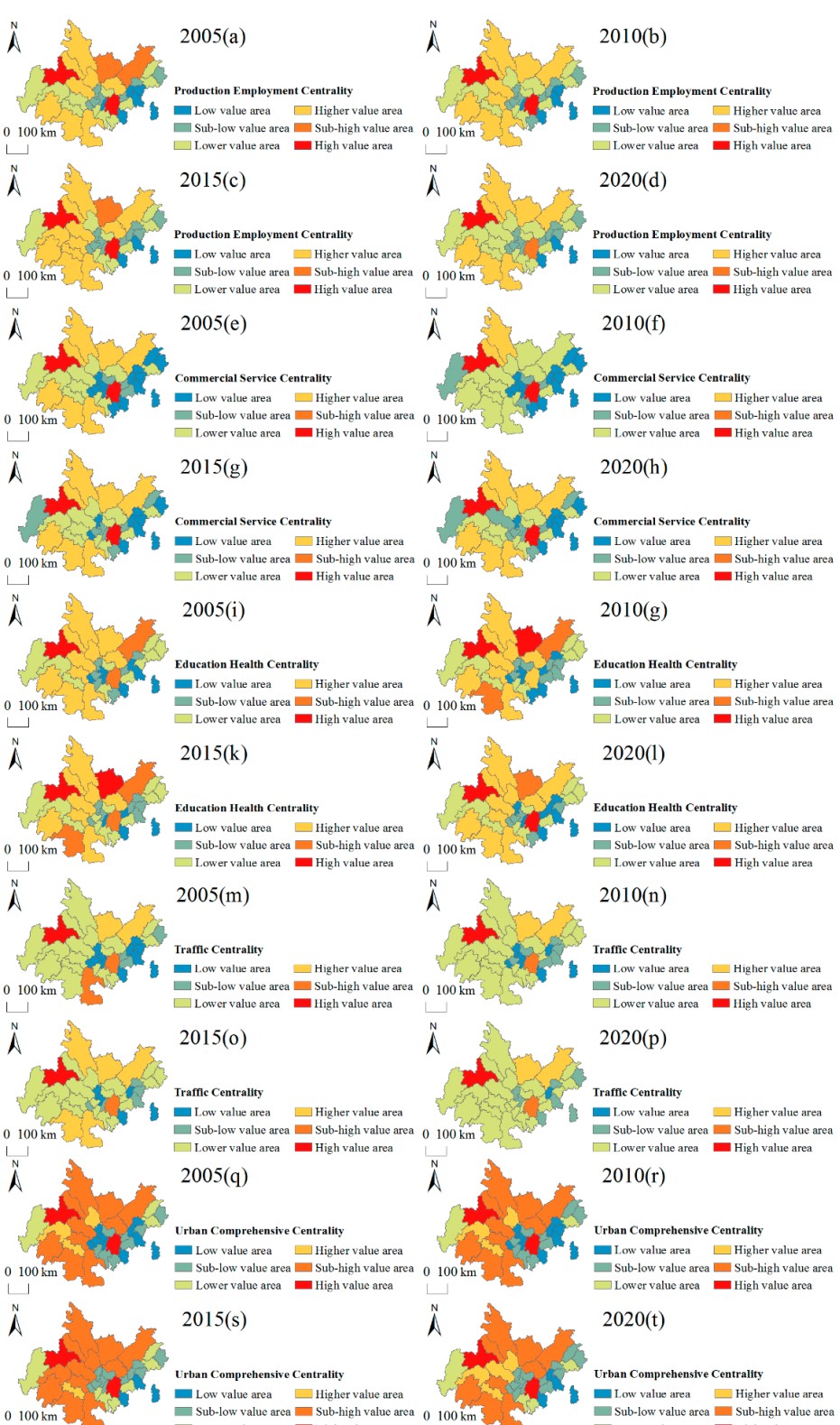

**Figure 2.** The centrality of the Chengdu-Chongqing urban agglomeration in different years. Pictures (**a**–**d**) represent the centrality of production employment, (**e**–**h**) represent the centrality of commercial service, (**i**–**l**) represent the centrality of education health, (**m**–**p**) represent the centrality of traffic, (**q**–**t**) represent the centrality of urban comprehensive.

*3.1. Analysis of the Centrality of Chengdu-Chongqing Urban Agglomeration*

As can be seen from Figure 2a–d, in 2005, the production and employment centers of the Chengdu-Chongqing urban agglomeration were dominated by the central urban areas of Chengdu and Chongqing, which were within the high-value zone. The overall production and employment centrality of the Chengdu urban agglomeration was higher than that of the Chongqing urban agglomeration; most of the centrality of the cities around Chengdu was located in sub-high-value areas and high-value areas, and the distribution centered on Nanchong and Dazhou was also more prominent, ranking only second to the central urban areas of Chengdu and Chongqing, while four cities—Ya'an, Meishan, Neijiang and Zigong—are in the lower-value area. The production and employment centrality of the cities around the central city of Chongqing is mainly in the low-value area, the sub-low-value area and the lower-value area. Taking the central urban area of Chongqing as the core, it decreases in a "circle-layer" manner. By 2020, the only high-value area for production and employment will be Chengdu. The central urban area of Chongqing has dropped from a high-value area to a sub-high-value area. The lower-value areas of the Chengdu urban agglomeration increased. The imbalance of production and employment has increased. The centrality of production and employment in the Chengdu-Chongqing urban agglomeration has formed a spatial pattern dominated by Chengdu and the central urban area of the Chongqing sub-center. Chengdu mainly develops the primary industry and the secondary industry and has a comparative advantage in these two industries. Due to the high urbanization rate in the central city of Chongqing, it has a comparative advantage in the tertiary industry. Therefore, we can focus on the development of primary industries such as agriculture, forestry, animal husbandry and fishery and secondary industries such as industry in Chengdu and focus on the development of tertiary industries such as tourism and catering in Chongqing, so as to strengthen the dual core advantages of Chengdu and Chongqing.

As can be seen from Figure 2e–h, in 2005, the commercial service center of the Chengdu-Chongqing urban agglomeration was dominated by the central urban areas of Chengdu and Chongqing, which were in the high-value area. The overall commercial service center of the Chengdu urban agglomeration was higher than that of the Chongqing urban agglomeration, with more high-value areas in the south and north and mostly lower and lower-value areas in the center. By 2020, the high-value areas of trade service centrality are still dominated by the central urban areas of Chengdu and Chongqing, showing a good concentration effect of trade services; the central urban areas are dominated by lower-value areas, sub-low-value areas, as well as low-value areas; the cities in the north and south are dominated by higher-value areas; the east is mainly dominated by low-value areas; the trade service centrality of the Chengdu-Chongqing urban agglomeration has formed a pattern dominated by the twin cores of the central urban areas of Chengdu and Chongqing. Chengdu and Chongqing are both important commercial and financial centers in Southwest China and have great resource endowment advantages. In the future, it is necessary to continue to promote the development of business services, high-tech services and scientific and technological services, strengthen service capabilities, enhance the radiation force of the two core cities, and promote the common development of the Chengdu-Chongqing urban agglomeration.

As can be seen from Figure 2i–l, in 2005, high-value areas of education and health centrality in the Chengdu-Chongqing urban agglomeration were dominated by Chengdu City. The sub-high-value area was distributed in the central cities of Chongqing and Dazhou. The higher-value areas of education and health centrality were mainly distributed in the Chengdu urban agglomeration. The low-value areas, the second low-value areas, and the lower-value areas were mainly distributed in the Chongqing urban agglomeration. In 2020, the central cities of Chongqing and Chengdu are in the high-value area, but the center of the surrounding cities is still dominated by low-value areas and sub-low-value areas, and the imbalance of education and health level among regions has intensified. Nanchong City has replaced Dazhou City as the only sub-high-value area. The education and health center

of Chengdu-Chongqing urban agglomeration has formed a spatial pattern with Chengdu and Chongqing as the leading cities and Nanchong as the deputy center. Therefore, it is necessary to expand basic public education and health services, co-ordinate the layout of educational and medical resources, break the boundaries between cities, and take the lead in realizing co-urbanization in Chongqing metropolitan area and Chengdu metropolitan area. The number of licensed doctors and the number of beds in health institutions in the central urban areas of Chengdu and Chongqing accounted for 38.67% and 31.11% of the entire urban agglomeration, and the number of full-time teachers in ordinary middle schools accounted for 22.84%. Both the levels of medical and health services and education and culture are much higher than other cities. Therefore, optimizing the allocation of public resources and promoting the equalization of the supply of basic public service resources are the keys to narrowing the development gap of the Chengdu-Chongqing urban agglomeration.

As can be seen from Figure 2m–p, in 2005, the high-value area of traffic centrality of the Chengdu-Chongqing urban agglomeration was distributed in Chengdu, the sub-high-value area was distributed in the central city of Chongqing and Luzhou, the higher-value area was distributed in Nanchong and Dazhou, and the overall traffic centrality of the Chengdu-Chongqing urban agglomeration was mainly in the lower-value area, while the low-value area and the sub-low-value area were mainly distributed in the Chongqing urban agglomeration. The unevenness of traffic development was remarkable. In 2020, the traffic centrality of the Chengdu-Chongqing urban agglomeration will still be dominated by sub-low-value areas, and the traffic centrality structure of Nanchong and Dazhou will also be prominent. Low-value and sub-low-value areas are still distributed around the central urban area of Chongqing. The transportation centrality of the Cheng-du-Chongqing urban agglomeration has formed a spatial pattern with Chengdu leading and the central urban area of Chongqing as the sub-center. The total road mileage in the main urban areas of Chengdu and Chongqing accounts for 22.05% of the total, with strong accessibility and significant transportation advantages. Relying on the main trunk lines of Chengdu and Chongqing, building the "backbone" supporting the development of the Chengdu-Chongqing urban agglomeration is an important measure to radiate and drive the development of cities along the line.

As can be seen from Figure 2q–t, in 2005, the high-value areas of integrated centrality of the Chengdu-Chongqing urban agglomeration were mainly located in the central cities of Chengdu and Chongqing, followed by the sub-high and higher-value areas in the Chengdu urban agglomeration, while the low-value areas, the sub-low-value areas and the lower-value areas were mainly in the Chongqing urban agglomeration, with the urban agglomeration of Chengdu being more central than the Chongqing urban agglomeration. In 2015, the integrated centrality of the Chengdu-Chongqing urban agglomeration increased, with the integrated centrality of Meishan and Suining rising from the higher- to the sub-high-values, and the lower-value areas of the Chongqing urban agglomeration decreasing. The urban agglomeration of Chongqing has decreased in the lower-value areas. By 2020, the integrated centrality of the Chengdu-Chengdu urban agglomeration decreases, with the integrated centrality of Suining, Ziyang and Meishan dropping from high-value area to a higher-value area, the integrated centrality of Yongchuan, Fuling and Kaizhou Districts dropping from a lower-value area to a sub-low-value area, and Dianjiang District county dropping from a sub-low-value area to a low-value area. The high-value areas are still distributed in the central urban areas of Chengdu and Chongqing. The comprehensive centrality of the Chengdu-Chongqing urban agglomeration is still dominated by the spatial pattern of the central urban areas of Chengdu and Chongqing. Chengdu is a vice-provincial national central city, Chongqing is a municipality directly under the Central Government of China, and the main urban area of Chongqing is the core functional area of Chongqing. The regional GDPs of the two cities are among the top 10 in the country and have strong economic strength. In terms of talent reserve, Chengdu has 879,300 undergraduate students in 2020, ranking fourth in the country. Chongqing's main urban area ranks seventh in

the country with 834,900 undergraduates. The scientific and technological innovation ability and talent-saving capacity of the dual core cities occupy great advantages. The imbalance of regional development level is significant. Therefore, it is necessary to enhance the core driving role of polar cities, fully demonstrate the polarization effect, and achieve coordinated development.

*3.2. Analysis of the Economic Development Level of Each City in the Chengdu-Chongqing Urban Agglomeration*

In this paper, GDP per capita represents the economic development level of the Chengdu-Chongqing urban agglomeration and the spatial evolution characteristics of the economic development level of the Chengdu-Chongqing urban agglomeration illustrated by the changes in GDP per capita in four different years. The GDP per capita of each city/region in the Chengdu-Chongqing urban agglomeration from 2005 to 2020 is visualized in ArcGIS 10.7 using the Natural breakpoint method, as shown in Figure 3.

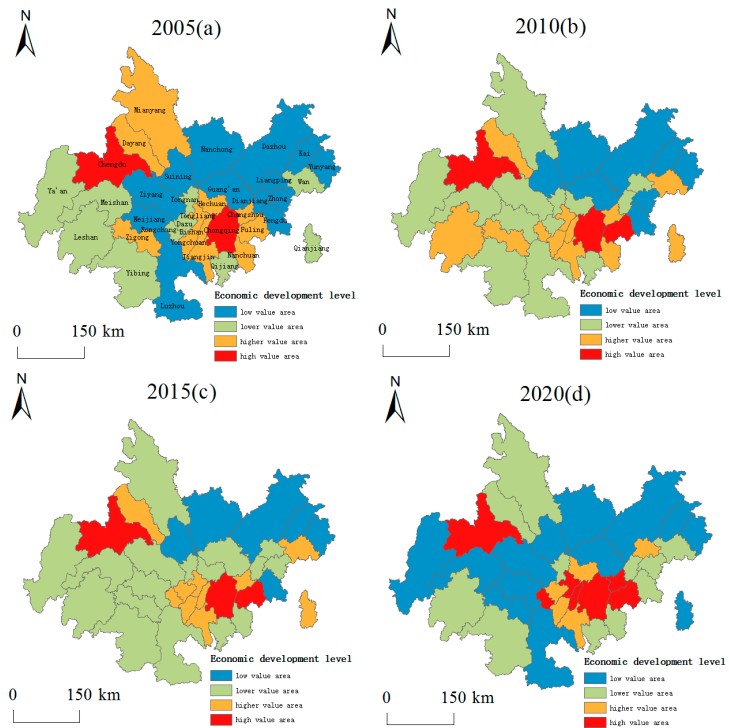

**Figure 3.** Spatial pattern of economic development levels in the Chengdu-Chongqing urban agglomeration in 2005 (**a**), 2010 (**b**), 2015 (**c**), and 2020 (**d**).

In 2005, among the 36 research units in the Chengdu-Chongqing urban agglomeration, the high-value areas of economic development were mainly located in the central urban areas of Chengdu and Chongqing, while the low-value areas were mainly located in the eastern part of the Chengdu urban agglomeration and the northern part of the Chongqing urban agglomeration. The economic development level of the Chongqing urban agglomeration shows a "circling" structure with the central urban area of Chongqing as the main focus, which shows that the regional development level of the Chengdu-Chongqing urban agglomeration is unbalanced, with Chengdu, Deyang, Mianyang, Zigong and the cities around the central urban area of Chongqing being relatively economically developed, while the southwest and northeast areas of Chengdu-Chongqing urban agglomeration are lagging behind and relatively economically backward. In 2010, the overall development level of the Chengdu-Chongqing urban agglomeration improved. The number of low-value areas decreased, with Mianyang City dropping from a higher-value area to a lower-value area, Leshan City rising from a lower-value area to a higher-value area, Ziyang, Neijiang, and Luzhou all rising from a low-value area to a lower-value area, Rongchang District,

Wanzhou District, and Qianjiang District rising from a lower-value area to a higher-value area, Fuling District rising from a higher-value area to a high-value area, and Hechuan District rising from a higher-value area to a high-value area. In 2015, the overall development level of the Chengdu-Chongqing urban agglomeration declined, with low-value areas mainly in northeastern Chengdu-Chongqing, lower-value areas mainly in southwestern and central Chengdu-Chongqing, and high-value areas in central Chengdu, Chongqing and Fuling. In 2020, in the Chengdu urban agglomeration, the core position of Chengdu became more prominent, and the lower-value areas of economic development level increased significantly, mainly distributed in the northeast and central part of Chengdu and Chongqing. In the Chongqing urban agglomeration, the central urban area, Bishan District, Changshou District, Tongliang District, Rongchang District, and Fuling District are within the high-value areas. Liangping District, Dazu District, Yongchuan District, Hechuan District, and Jiangjin District are within the higher-value areas. From an overall perspective, the Chongqing urban agglomeration has a more advanced level of economic development than the Chengdu urban agglomeration. The Chengdu-Chongqing urban agglomeration still has a relatively obvious problem of economic development differences. Therefore, the cities of the Chengdu-Chongqing urban agglomeration should break down administrative barriers and deepen inter-governmental cooperation and exchanges. At the same time, it is necessary for Chengdu and Chongqing to strengthen policy exchanges in the core functional areas of the urban agglomeration to promote the economic development of the peripheral cities of the Chengdu-Chongqing urban agglomeration and the cities bordering Chengdu and Chongqing. Also, give full play to the role of a nuclear city, radiate farther, and promote the coordinated development of cities in the Chengdu-Chongqing urban agglomeration.

## 4. Analysis of the Intensity of Economic Linkages of Chengdu-Chongqing Urban Agglomeration

### 4.1. Analysis of the Linkages Volume and the Proportion of the Linkages Volume in the Chengdu-Chongqing Urban Agglomeration

Using the gravity model, the total economic linkages of cities within the Chengdu-Chongqing urban agglomeration in 2005, 2010, 2015, and 2020 were calculated.

From the economic linkages of cities in the four years (Figure 4), we can find that, during 2005–2020, the economic linkages of cities in the Chongqing urban agglomeration showed an overall trend of growth, with substantial growth in the central city of Chongqing, while the economic linkages of cities in the Chengdu urban agglomeration showed an overall trend of decline, with a larger decline in Ziyang, and smaller change in the economic linkages and the ratio of each city to the whole region. In 2005, the most closely linked cities in the Chengdu-Chongqing urban agglomeration were the Chengdu urban agglomeration, with Chengdu as its core. The cities most closely related to Chengdu were Leshan, Meishan, Ziyang, Deyang, and Mianyang, with these five cities accounting for 34.35% of the total economy, followed by the Chongqing urban agglomeration, with the central urban area of Chongqing as its core, including a total of 21 surrounding districts and counties, accounting for 28.04% of the economic ties. In 2010, the total economic linkages of the Chengdu city cluster, with Chengdu as the core, accounted for 30.39% of the region, a decrease compared to 2005, while the Chongqing city cluster accounted for 28.71%, a slight increase. In 2015, the total economic linkages accounted from the Chongqing city cluster was 32.80%, with the largest increase in economic linkages being in the central city of Chongqing, from 822.03 to 1228.67. The value of Chengdu city cluster was 30.53%, with little alteration. By 2020, the share of economic linkages between Chengdu and its neighboring cities tends to decrease, shrinking to 22.65%, leaving only Meishan, Deyang, and Mianyang as the three cities with the closest ties Chengdu. The uneven economic development within the Chengdu city cluster, with Chengdu as its core, has further intensified. The share of total economic linkages in the Chongqing city cluster has further increased, reaching 36.07%, and the unbalanced economic development within the city cluster has improved. However,

on the whole, the imbalance in economic development within the Chengdu-Chongqing urban agglomeration is still increasing.

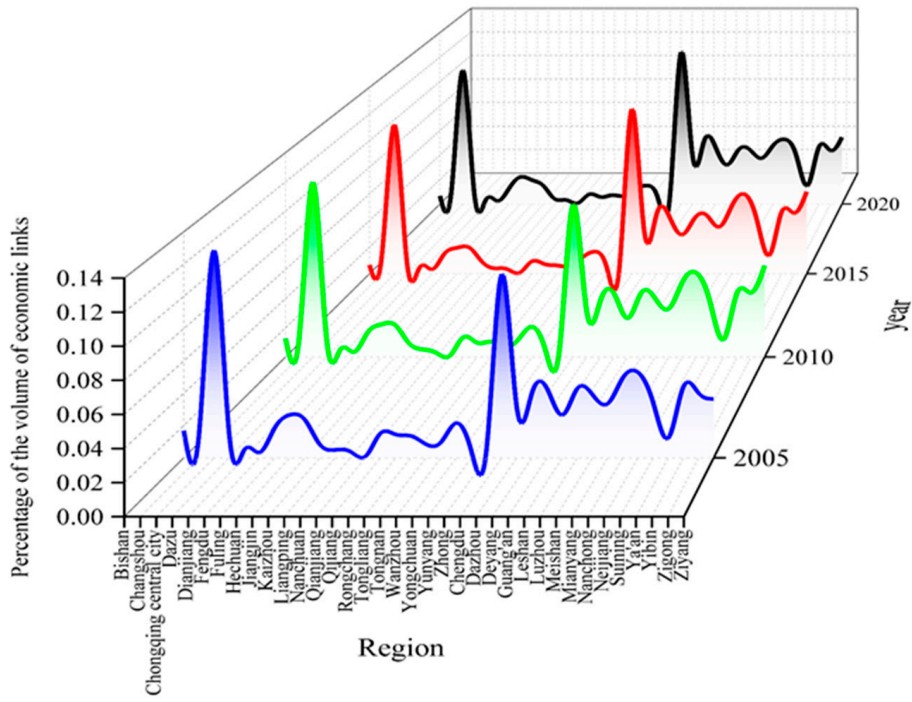

**Figure 4.** The percentage of the volume of economic links in each district in different years.

From 2005 to 2020, the evolution trend of the total economic connection of the Chengdu-Chongqing urban agglomeration is as follows: (1) The economic connection between Chengdu and the central urban area of Chongqing is the largest, and the spatial distribution of economic development is uneven; (2) The total economic connection of the Chongqing urban agglomeration with the central urban area of Chongqing as the core generally shows an upward trend, and the total economic connection of the Chengdu urban agglomeration with Chengdu as the core generally shows a downward trend. (3) The spatial change of the total urban economic connection. The characteristics are: From 2005 to 2020, the imbalance between Chengdu and surrounding cities intensified, and the imbalance between Chongqing and surrounding cities eased, but in general, the imbalance of regional economic development was still significant.

### 4.2. Analysis on the Spatial-Temporal Pattern of Linkages Intensity of Cities in Chengdu-Chongqing Urban Agglomeration

Using ArcGIS10.7 software to visualize the correlation intensity data in four time periods of 2005, 2010, 2015, and 2020, the spatial pattern map of the economic linkage of Chengdu-Chongqing urban agglomeration was obtained, and the cities' economic linkages intensity values were roughly divided into three intervals of 0.01~30, 30~71 and 71~175 according to the Natural breakpoint method, which can be combined with Figure 5. It is found that the linkages intensity of cities in the Chengdu-Chongqing urban agglomeration is weak during the period 2005–2020, and the gravitational value of each city is small, mainly concentrated in the interval of 0.01–30. In 2005, cities with high economic linkages intensity were Chengdu-Leshan, Chengdu-Meishan, Chengdu-Ziyang, Chengdu-Deyang, Chengdu-Mianyang, Deyang-Mianyang, Neijiang-Ziyang, Neijiang-Zigong, Yibin-Luzhou, and Chongqing Central City-Guang'an, with linkages intensity values in the range of 71 to 175. It can be seen that the linkages intensity between cities in the Chengdu city circle is high and dense, and several cities closer to Chengdu have relatively large gravitational values, while the economic linkages intensity between Chongqing city groups is weaker compared with that between Chengdu city groups, and

the linkages intensity between Chongqing Central City and the surrounding counties and districts is mostly in the range of 30 to 71. In 2010, the intensity of association between all cities decreased, with Chengdu and the central city of Chongqing as a whole being the two poles with the greatest intensity of association with their neighboring cities, with weaker interaction between cities in central and northeastern Chengdu and Chongqing, and weaker interaction between the Chongqing urban agglomeration and other cities in Sichuan province, with significant imbalances in development between regions. In 2015, the intensity of association between cities had increased to varying degrees, and a network development pattern with Chengdu and Chongqing central urban areas as the core has taken shape. However, the intensity of association between the Chengdu city cluster and the Chongqing city cluster is still weak, and the phenomenon of unbalanced regional development has not been greatly improved. By 2020, the number of cities closely related to each other in Chengdu and Chongqing will be significantly reduced, and the "double core" pattern, with Chengdu and Chongqing's central urban areas as the core, and Meishan, Deyang, and Mianyang as nodes in Sichuan, and Chongqing, with its central urban areas as the core, and Hechuan and Bishan as nodes in Chongqing, will be radiating. The intensity of association between cities in the central Chengdu-Chongqing region is weak, with the intensity of association mainly in the range of 0.01–30, showing a collapse of the economy in the central part of the country and an increase in the imbalance of development between regions.

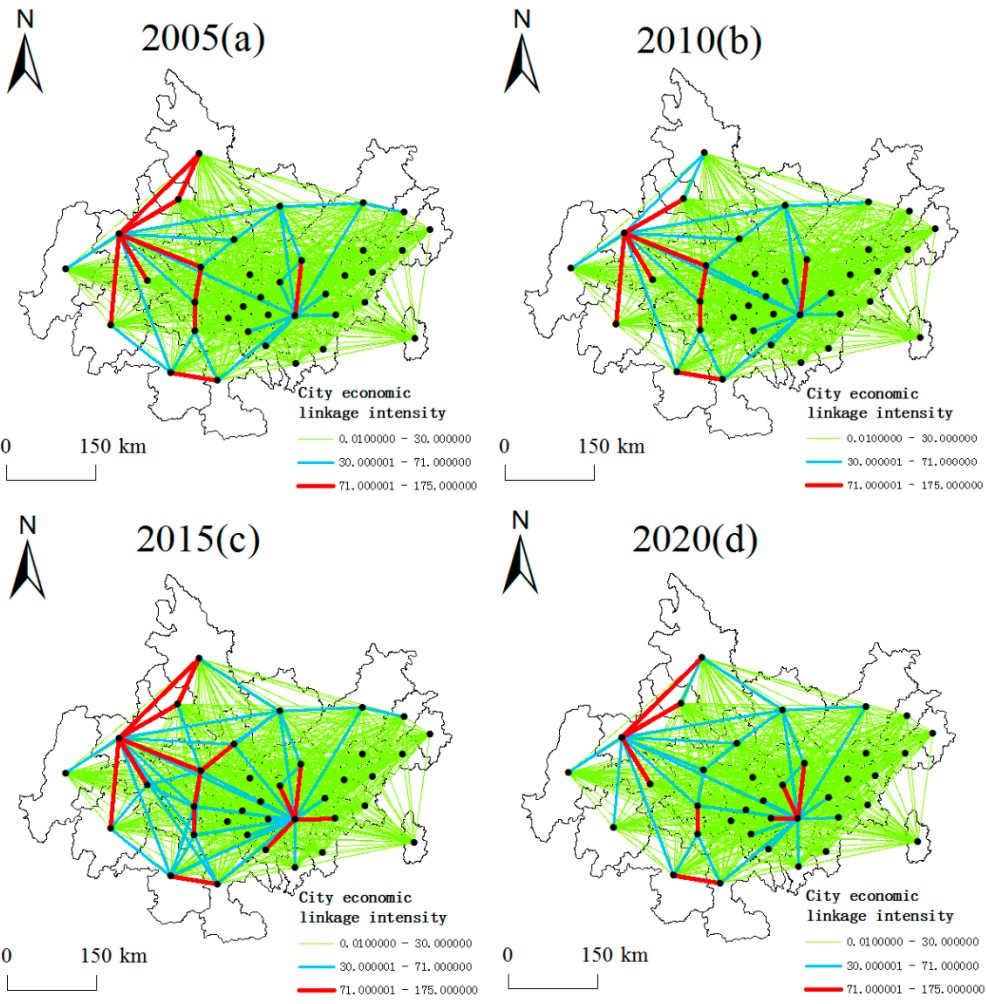

**Figure 5.** Economic linkages intensity of the Chengdu-Chongqing urban agglomeration in 2005 (**a**), 2010 (**b**), 2015 (**c**), and 2020 (**d**).

The changes in the pattern of linkages intensity among cities in the Chengdu-Chongqing urban agglomeration from 2005 to 2020 are as follows: (1) The area with the strongest economic interconnection among cities is in the Chengdu urban agglomeration with Chengdu as the core, followed by the area with stronger interconnection among cities in the Chongqing urban agglomeration with the central city of Chongqing as the core, while the linkages intensity among other cities is weaker, especially in the central region where the interconnection intensity values among cities are at low values, and the economic collapse is obvious. (2) The number of city pairs closely related to each other in the Chengdu urban agglomeration has decreased significantly, mainly concentrated in Chengdu and its surrounding cities, while the number of city pairs closely related to each other in the Chongqing urban agglomeration has increased, highlighting the "double core" structure led by Chengdu, and Chongqing central city, but with a weak external radiation effect.

## 5. Analysis of the Factors Influencing the Economic Development of the Chengdu-Chongqing Urban Agglomeration

### 5.1. Analysis of Impact Factor Geographic Detector Results

The economic development of the Chengdu-Chongqing urban agglomeration is affected and restricted by many factors. This paper uses per capita GDP to represent the explained variables to explore the influencing factors of economic development of Chengdu-Chongqing urban agglomeration from four aspects: production and employment, business services, education and health, and transportation. Taking into account the availability of data, select employees in urban non-private, rural employees, the total output value of the primary industry, the total output value of the secondary industry, total social fixed asset investment, total retail sales of social consumer goods, the total output value of the tertiary industry, number of ordinary primary and secondary schools, number of pupils in general primary and secondary schools, number of health institutions, number of beds in health institutions, total road mileage, road passenger turnover, and road freight turnover—a total of 14 indicators.

Using the geographic detector to detect the factors affecting the economic development of the Chengdu-Chongqing urban agglomeration from 2005 to 2020, the $q$-value was obtained. According to the factor detection results, the p-values of the 14 indicators have passed the 1% significance test in four time periods, indicating that the selected relevant influencing factors have a strong and significant impact on the development of the Chengdu-Chongqing urban agglomeration.

The heat map (Figure 6) shows the change of q value of different impact factors from 2005 to 2020, and the q value greater than 0.5 is marked in the figure. The intensity of the effects of different influencing factors on the economic development of the Chengdu-Chongqing urban agglomeration varies from year to year, but the $q$-values of the total output value of the secondary industry, total social fixed asset investment, number of beds in health institutions, and road freight turnover are all greater than 0.5 in all four years, with consistently strong explanatory ability. From 2005 to 2020, the explanatory ability of production and employment factors, business and trade service factors, and transportation factors for economic development showed a steady decline trend. Among them, urban non-private sector employees, total retail sales of social consumer goods, the total output value of the tertiary industry, and road passenger turnover in 2020 have $q$-values less than 0.5, and the influence is relatively weak compared with previous years. Among the factors of education and health, the explanatory ability of health level to economic development also shows a downward trend, while the level of education shows a steady increase trend, indicating that the impact of education on economic development is becoming more and more important. In 2005, the effect of business and trade services on economic development was more significant than other factors. Among them, the output value of the tertiary industry had the most significant impact, with a $q$-value around 0.8195. From 2010 to 2015, the effect of traffic factors on economic development exceeded other factors, which indicates that the relationship between traffic development and economic development is

closer during this period. Road freight turnover has the most significant explanatory ability for economic development in 2010 and 2015, with *q*-values of about 0.6992 and 0.7316, respectively. By 2020, education level has become a significant factor affecting economic development. The number of ordinary primary and secondary schools has the strongest explanatory ability among all detection factors, with a *q*-value of 0.5896.

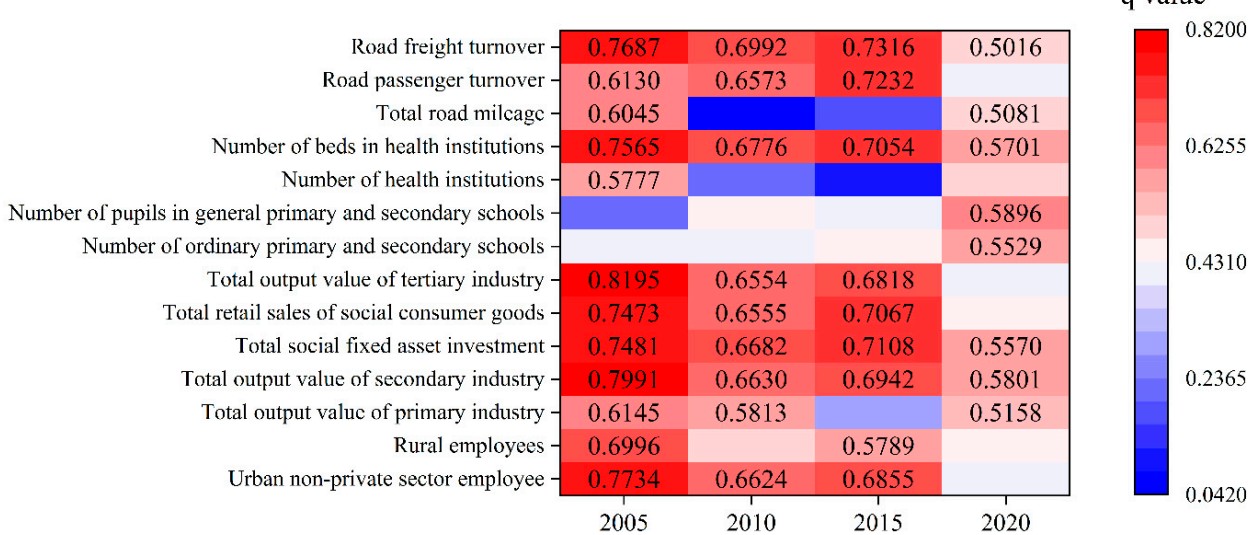

**Figure 6.** Heatmap of *q*-values for each factor in different years.

The total output value of the secondary industry, total social fixed asset investment, number of beds in health institutions, and road freight turnover are the indicators with consistently strong explanatory ability for economic development, while the *q*-values of the other indicators are all less than 0.5 in the period 2005–2020, with relatively unstable explanatory ability.

### 5.2. Analysis of Spatial Variation in the Role of Influencing Factors

The Moran's I values and statistical test Z values were calculated for 2005, 2010, 2015, and 2020 (as shown in Table 1), separately. Further, 2005 and 2020 Moran's I and Z values passed the 1% statistical test. The Moran's I and Z values for 2010 and 2015 passed the 10% statistical test, indicating the existence of a positive spatial autocorrelation, and the Moran's I values for the four years were in the interval of [0.1192, 0.4198], with apparent clustering effects.

**Table 1.** Moran's I, Z-value, and *p*-value test results.

| Year | Moran's I | Z-Value | *p*-Value |
|------|-----------|---------|-----------|
| 2005 | 0.2825 | 3.8256 | 0.0001 |
| 2010 | 0.1192 | 1.7846 | 0.0743 |
| 2015 | 0.1210 | 1.8131 | 0.0698 |
| 2020 | 0.4198 | 5.2504 | 0.0000 |

According to the analysis of geographical detector results, the factors with *q*-values greater than 0.5 in the four years are the total output value of the secondary industry, total social fixed asset investment, number of beds in health institutions, and road freight turnover, which have strong and sustainable explanatory ability for the economic development of Chengdu-Chongqing urban agglomeration. Therefore, these four significant influencing factors are selected for GWR local spatial regression analysis to explore the spatial differences of the action direction and intensity of the four significant factors in different research units in the four time periods. The results in Table 2 show that the corrected $R^2$ of the model is 0.666, 0.651, 0.760, and 0.766 in 2005, 2010, 2015, and 2020

respectively, indicating that the fitting effect of the model is good. The condition number is greater than 0 and less than 30 in four time periods, indicating that the model has passed the multicollinearity test, the calculation result of the model is reliable, and the result of the factor detector is also reliable. The effects of each variable are spatially non-stationary, but the degree of difference and characteristics is different (Figure 7).

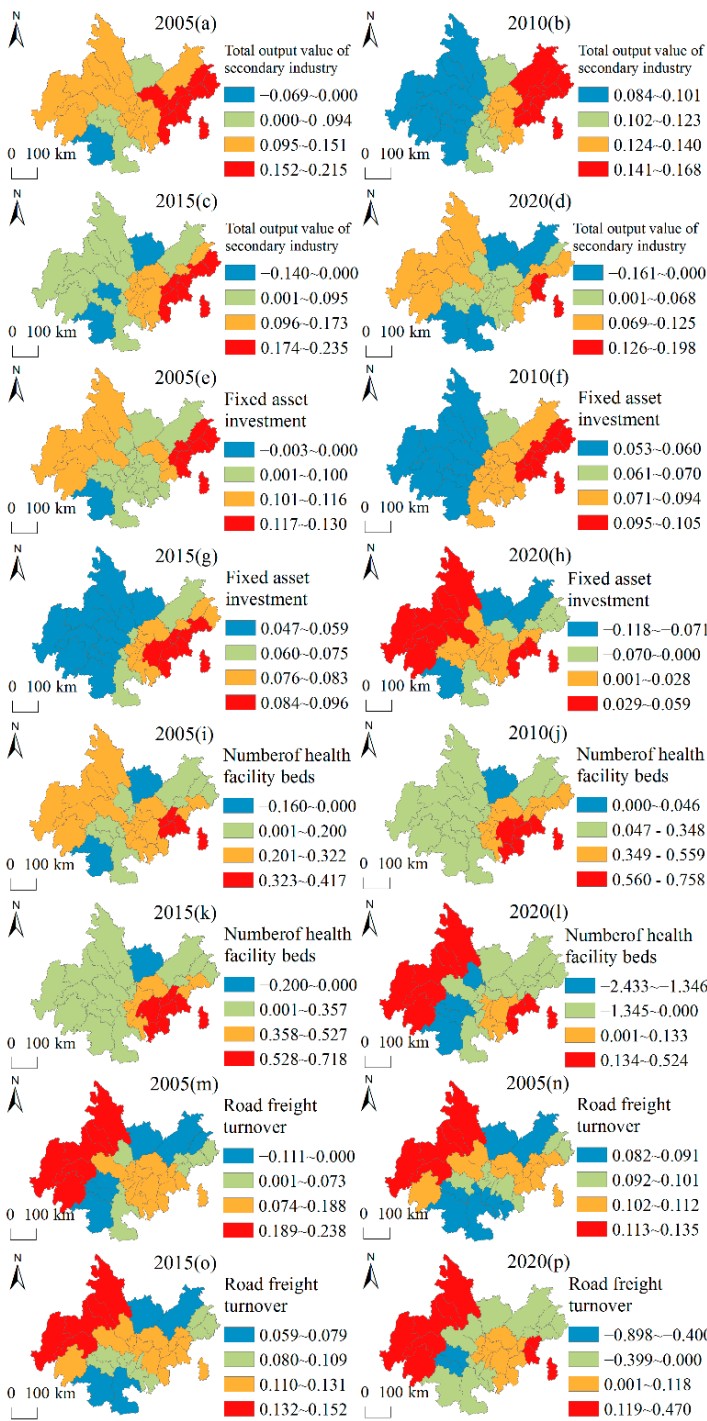

**Figure 7.** Four-year comparison of regression coefficients for significant impact factors. Pictures (**a–d**) represent the regression coefficient of total output value of secondary industry, (**e–h**) represent the regression coefficient of fixed asset investment, (**i–l**) represent the regression coefficient of number of health facility beds, (**m–p**) represent the regression coefficient of road freight turnover.

**Table 2.** Statistical test results.

| Year | 2005 | 2010 | 2015 | 2020 |
|------|------|------|------|------|
| Model goodness of fit | 0.730 | 0.733 | 0.795 | 0.793 |
| Calibration model goodness of fit | 0.666 | 0.651 | 0.760 | 0.766 |
| Partial $R^2$ | 0.566~0.793 | 0.617~0.841 | 0.759~0.792 | 0.792~0.793 |
| Conditions | 14.869~29.993 | 21.268~29.990 | 25.982~29.997 | 23.803~23.829 |

(1) The total output value of the secondary industry is positively correlated with the dependent variable in all units of analysis except for 2010. During 2005–2020, the positively correlated areas are mainly located in the eastern and western regions of Chengdu-Chongqing urban agglomeration, while the negatively correlated areas are located primarily in the northern and southern regions. The overall regression coefficient shows the characteristics of "high in the east and west, low in the north and south," and the absolute value of the regression coefficient of the positive correlation areas decreases. It shows that the promotion effect of the output value of the secondary industry on economic development decreases, the negative correlation area increases, the absolute value of the regression coefficient increases, and the inhibition effect increases. From the time-space evolution process of the output value of the secondary industry, after 15 years of development, the cities in the east and west of the Chengdu-Chongqing urban agglomeration promote economic development, but the promotion effect is weakened. The output value of the secondary industry in the cities of the north and south is negatively correlated with the dependent variable, and the inhibition effect is strengthened.

(2) The regression coefficients on total social fixed asset investment are much less volatile than the other three variables. The effects of the regression coefficients vary less spatially, with a positive correlation with the dependent variable in 2010 and 2015, a slightly weaker correlation for the east and west regions of Chengdu-Chongqing urban agglomeration, and a slightly stronger correlation for the central north and south regions. In 2005, the center of the high value of the regression coefficient was concentrated in the northeastern parts of Chengdu-Chongqing urban agglomeration, with a greater dependence on the amount of fixed asset investment, resulting in a higher regression coefficient. By 2020, with the increase in total social fixed asset investment, the high-value area of the regression coefficient tends to shift to the west, and the value of the coefficient decreases, but is still positive, reflecting the driving effect of Chengdu on the surrounding cities. The regression coefficient of the cities in the north and south of Chengdu-Chongqing urban agglomeration is negative, indicating that the increase in the amount of fixed asset investment will instead inhibit the economic growth of these areas.

(3) The regression coefficient for the number of beds in health institutions is positive in all units of analysis only in 2010, with a significant suppressive effect on Nanchong and Yibin in 2005 and a positive promotional effect on all other cities. By 2020, the positive values of the regression coefficient are distributed in the eastern and western regions of the Chengdu-Chongqing urban agglomeration, and the negative values are distributed in the northeastern and southwestern axial zones. As the number of beds in health institutions increases, their contribution to the economically backward regions of the eastern part of the Chengdu-Chongqing urban agglomeration is obvious, and their economic contribution to the Chengdu urban agglomeration with Chengdu at its core is gradually becoming significant.

(4) The regression coefficient of road freight turnover fluctuates slightly in the first three years. In 2020, the spatial difference of its effect was significant, and the regression coefficient is around −0.899~0.469, but it is a mainly positive correlation. The positive area is concentrated in the west of Chengdu-Chongqing urban agglomeration and the Chongqing-urban agglomeration, with the central urban area of Chongqing as the core. At the same time, its effect is negative in the cities in the north and south of the Chengdu-Chongqing urban agglomeration. It shows that the transportation infrastructure in these

areas is not perfect, and the role of transportation in promoting the economy has not been brought into full play.

In general, the promoting effect of the above four factors on economic development is mainly concentrated in the eastern and western regions of the Chongqing-urban agglomeration. It mainly inhibits the cities in the north and south. The most significant difference in the spatial effect of road freight turnover indicates that the uneven development of the transportation network in the Chengdu-Chongqing urban agglomeration is significant. In the relevant construction affecting economic development, attention should be paid to the supporting role of the transportation network.

## 6. Conclusions and Recommendations

### 6.1. Main Conclusions

This paper constructs 14 evaluation index systems based on four subsystems of production and employment, trade and commerce services, education and health, and transportation in 36 cities of the Chengdu-Chongqing urban agglomeration. Using GDP per capita to represent the level of economic development, the evolution characteristics of the economic spatial pattern of the Chengdu-Chongqing urban agglomeration and its influencing factors are explored based on the performance of feedback effects in four time sections from 2005 to 2020, with the help of gravity models, geographic detector, geographically weighted regression, and other methods. The following main conclusions were drawn.

(1) From the evolution of the integrated centrality of Chengdu-Chongqing urban agglomeration, the high-value area of integrated centrality is a spatial pattern dominated by Chengdu and Chongqing central cities, with a significant agglomeration effect. The overall centrality of the Chengdu urban agglomeration is higher than that of the Chongqing urban agglomeration, and the uneven development between regions is still relatively obvious.

(2) The economically developed areas of the Chengdu-Chongqing urban agglomeration are mainly concentrated in Chengdu and Chongqing central cities and their surrounding areas. The economic development level of the neighboring districts and counties in the Chongqing central city is higher than that of the surrounding areas of Chengdu, indicating that the central city of Chongqing has a strong economic drive and significant radiation capacity to the neighboring regions.

(3) In terms of the urban economic linkages volume, the Chengdu urban agglomeration and the Chongqing urban agglomeration account for a significant proportion of the economic linkages, while economic links between other cities are weaker, especially in the central region where the collapse situation is obvious. On the whole, a "dual-core" pattern has been formed, with the central cities of Chengdu and Chongqing as the leading cities.

(4) In terms of the driving factors affecting economic development, the total output value of the secondary industry, total social fixed asset investment, number of beds in health institutions, and road freight turnover have consistently strong explanatory abilities for economic development. The four significant factors mostly promote the economic development of most cities in Chengdu-Chongqing urban agglomeration and inhibit a small number of cities. The promotion effect is mostly concentrated in the eastern and western cities of Chengdu-Chongqing urban agglomeration, and the inhibition effect is mostly concentrated in the southern and northern cities of Chengdu-Chongqing urban agglomeration.

### 6.2. Recommendations

Based on the above findings, the following recommendations are made to promote the collaborative economic development of the Chengdu-Chongqing urban agglomeration.

(1) Establish an open, competitive, and orderly market system for the city cluster. Break down administrative barriers between the cities in the Chengdu-Chongqing urban agglomeration. Promote the formation of a pattern of complementary resource advantages, rational division of functions, and infrastructure interconnection among cities. Give full

play to the strategic supporting role of the Yangtze River Economic Belt and expand new space for development. Increase the economic ties between the northeastern and southwestern cities of the Chengdu-Chongqing urban agglomeration and Chengdu and Chongqing. At the same time, attention should be paid to the economic collapse of the central part of the Chengdu-Chongqing urban agglomeration to prevent severe regional differences in development.

(2) Enhance the radiation effects of the two core cities of Chengdu and Chongqing. Strengthen the important functions of production and employment, business services, education and health, and transportation hubs in the Chengdu-Chongqing urban agglomeration. Give full play to the core driving ability of Chengdu and speed up the process of urbanization in surrounding cities such as Deyang, Ya'an, Meishan, and Ziyang. Strengthen the strategic support of the western development of Chongqing Metropolis and the carrier function of the central hub in the west part of the Yangtze River Economic Belt. Take the central urban area of Chongqing as the core, link the urban belt along the river and the adjacent cities in Sichuan, and promote regional coordinated development.

(3) Pay attention to the role of significant driving factors. Improve the core competitiveness of industrial enterprises, mobilize the enthusiasm of private investment, enhance the service capabilities of primary health care institutions, and speed up the construction of road transport infrastructure. Strengthen the supporting role of production, commerce, health, and transportation networks for the economic development of the Chengdu-Chongqing urban agglomeration.

**Author Contributions:** Conceptualization, R.D.; methodology, R.D. and J.F.; validation, J.F., Y.Z. and T.Z. (Tao Zhou); formal analysis, J.F. and T.Z. (Tao Zhou); investigation, J.F.; resources, J.Y., T.Z. (Ting Zhang) and Y.D.; data curation, L.D.; writing—original draft preparation, R.D. and J.F.; writing—review and editing, R.D.; visualization, R.D. and J.F. All authors have read and agreed to the published version of the manuscript.

**Funding:** This work was supported by the National Natural Science Foundation of China (No. 72001053).

**Data Availability Statement:** The data presented in this study are available on request from the corresponding author.

**Acknowledgments:** We thank the editors and the anonymous reviewers for their valuable comments and suggestions.

**Conflicts of Interest:** The authors declare no conflict of interest.

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
