# Peer review of "Research on the Evolution of the Economic Spatial Pattern of Urban Agglomeration and Its Influencing Factors, Evidence from the Chengdu-Chongqing Urban Agglomeration of China"

_sustainability, doi:10.3390/su141710969_

Round 1

Reviewer 1 Report

The paper overall looks good.

Author Response

Point: The paper overall looks good.

Response: Thank you very much for your valuable comments and revise decision, we will continue to work hard to do it well.

Author Response

This paper analyzes the influencing factors on the agglomeration of a certain area of China utilizing several data sources. However, the contribution of this study is very limited. this Although it is said that the study combined temporal and spatial scales, the analysis only compared four time periods’ data and there isn’t any temporal scale incorporated in the methodology. I would also like to offer the following comments:

Response: Thank you for your valuable suggestions, and this kindly revise decision. Based on your meaningful comments, a major modification was done in the new version.

Point 1: It is wordy and repetitious. It is required to explain the research work with much less text.

Response 1: The comment is right. Our initial attempt to analyze the trends in results in each subsection of the evidence in detail led to lengthy repetition. Therefore, we deleted some less important analysis processes in subsection 3. For example, in the first four centrality analyses in subsection 3.1, we deleted the analysis processes in 2010 and 2015, and focused on analyzing the spatial pattern characteristics of each centrality in the first and last years. Detailed analysis of other empirical subsections we still think is necessary.

Point 2: In Table 1, locations and characteristics of areas should be presented instead of names of provincials and municipal districts. The map of the study area and the characteristics of the area should be explained first.

Response 2: This comment is extremely helpful. We did ignore the map representation and regional characteristics of the research area. Therefore, in Section 2.1, we replaced Table 1 with the distribution map of the research area and per capita GDP, and added a description of the area, population, and economic characteristics of the research area in the text.

As “Chengdu-Chongqing urban agglomeration is located in southwest China, with a total area of about 185,000 km2, with a resident population of 100.7 million in 2020, accounting for 6.9% of the country. The regional GDP is 6.8 trillion, ac-counting for 6.7% of the country. It is one of the regions with the best economic foundation and the strongest economic strength intensity in the west. It has the regional advantage of connecting the east and the west, and connecting the north and the south. At the same time, it has an excellent endowment of natural resources, strong comprehensive carrying capacity, strong foundation of manufacturing business, finance, and other industries, high degree of openness, rich human resources, and good innovation and entrepreneurship environment. It is a typical region with national economic importance and strong network connection characteristics.

Figure 1. Research area and GDP per capita distribution”

Point 3: There is no originality in the methodology but the description on the methodology is too long. So the explanation of models can be simplified with reference to existing literature and books.

Response 3: Thank you very much for your wonderful suggestion. We noticed that the introduction to the methodology was indeed too long, so we have shortened the description of the methods section in Section 2.3. In the analysis of influencing factors, we believe that the combination of geographic detectors and GWR models is still innovative.

Point 4: Figure 1 and 4 are too small to read.

Response 4: Thanks for this suggestion. We have re-done the empirical diagram, which has now been revised as Figure 2 and Figure 5, so that it can be read more clearly. Shown as Figure 2 on page 15 and Figure 5 on page 26.

Point 5: In section 3, it’s hard to catch the key point of findings. It is not clear how the authors define the range of low-value and high-value. The analysis results can be summarized with a numerical table at least to show which area is dominant and differences between areas by descriptive statistics of the centrality results. When interpreting the analysis results, the authors should explain the academic findings not only describing data itself. Why it is happening and how they can be used for the future measures. The same to other sections.

Response 5: Thanks a lot for your great revision suggestions. We have made a major revision in section 3. At the beginning of section 3, it is explained that each centrality is divided into 6 levels by the Natural breakpoint method in ArcGIS, and the academic interpretation of the results and for future measures, see pages 11 to 16 for detailed revisions.

Point 6: Table 2 is also hard to read. What is the objective of this data analysis and what the authors want the readers to read? The authors should modify tables and figures to be easily understood not only list up the data. The same to Table 3.

Response 6: Thank you for your valued suggestions. Table 2 lists in detail the economic connections of the 36 districts and counties in the Chengdu-Chongqing urban agglomeration from 2005 to 2020 and their proportions in the whole region, so as to more clearly show the temporal trends and gaps in the economic connections between regions. The analysis section at the end of subsection 4.1 also summarizes the evolution trend of economic linkages. Table 3 shows the four-year q value of the impact factor detected by the geographic detector. The q value represents the explanatory power of the independent variable to the dependent variable. The purpose is to screen out the factors with higher explanatory power for the level of economic development, and then use GWR to do an empirical analysis on the selected factors to analyze the direction and strength of their effects.

Reviewer 3 Report

Dear authors, 

Your paper entitled Research on the evolution of the economic spatial pattern of urban agglomeration and its influencing factors, evidence from the Chengdu-Chongqing urban agglomeration of China  it is quite interesting and it has a relevant contribution to scholarship. I appreciated the originality, the high quality of the structure and the clarity. Also, I saw a logical coherence in all the paper and the arguments used are very well based. 

Although, I suggest you to use like references mostly academic papers published in the last 1-3 years in journals indexed in SCOPUS and/or WoS/CA. In the same time, it will be good for you to use more references from the articles already published in MDPI.

I wish you to continue your valuable  research in the future.

Author Response

Point: I suggest you to use like references mostly academic papers published in the last 1-3 years in journals indexed in SCOPUS and/or WoS/CA. In the same time, it will be good for you to use more references from the articles already published in MDPI.

Response: Thank you for your valuable suggestions. Based on your suggestions, we have updated related new references 1 and 2 from MDPI journals, and reference 25. Most of the citations in the whole article are from MDPI journals, so we keep the citations of other references.

1.    Chen, Y.; Miao, Q.; Zhou, Q. Spatiotemporal Differentiation and Driving Force        Analysis of the High-Quality Development of Urban Agglomerations along            the Yellow River Basin. International Journal of Environmental Research and          Public Health. 2022, 19, 2484, doi:10.3390/ijerph19042484.

2.     Yu, S.; Kim, D. Changes in Regional Economic Resilience after the 2008                  Global Economic Crisis: The Case of Korea. Sustainability. 2021, 13, 11392,            doi:10.3390/su132011392.

25.   Wang, J.F.; Xu C.D. Geographic detector: principle and prospect. Acta                      Geographica Sinica. 2017, 72, 19, doi:10.11821/dlxb201701010.

Round 2

Reviewer 2 Report

1)        Authors do not need to remain revision history. It’s very hard to read the revised version. Please reflect all revisions and highlight revised part with colors.

2)        It is good to add the map and GDP levels in Fig.1. But the legend of GDP levels can be modified into round numbers such as 30000-40000. Also, the unit of GDP can be 1000CNY.

3)        Why the authors chose the natural breakpoint method? Is it Natural breakpoint with or without Jenks Optimization? There are several standard classification methods in ArcGIS, such as Equal Interval, Quantile, and Standard Deviation. Since the authors observes the area differences according to this classification, the applicability and advantages of the classification method should be discussed. Also, the reason to set six classes should be discussed. Why there are six classes in Fig. 2 but four classes in Fig. 3? Do the authors test different number of classes? The name of each class is also quite strange. What is the difference between low and lower? Why do not name them clearer way such as low, medium, high, superhigh…? What are the criteria between classes? These issues should be clearly explained.

4)        It is still very hard to understand Table 2 and 3. It will be good to make graphs to show the differences between areas and years instead of Table 2 and 3. For the name of region it will be good to indicate which class (classes in fig. 2 and 3) they belong to. It is recommended to categorize or order them according to the classes.

Author Response

Response to Reviewer 2

Comments

Point 1: Authors do not need to remain revision history. It’s very hard to read the revised version. Please reflect all revisions and highlight revised part with colors.

Response 1: The comment is right. Please forgive us for the inconvenience we have caused. As your suggestion, this new version will not keep the revision history and only highlights the revised part in color.

Point 2: It is good to add the map and GDP levels in Fig.1. But the legend of GDP levels can be modified into round numbers such as 30000-40000. Also, the unit of GDP can be 1000CNY.

Response 2: Thank you very much for your valuable comments. We have modified the label range of figure 1 to an integer representation and changed the unit as 1000yuan according as your requirements.

Point 3: Why the authors chose the natural breakpoint method? Is it Natural breakpoint with or without Jenks Optimization? There are several standard classification methods in ArcGIS, such as Equal Interval, Quantile, and Standard Deviation. Since the authors observes the area differences according to this classification, the applicability and advantages of the classification method should be discussed. Also, the reason to set six classes should be discussed. Why there are six classes in Fig. 2 but four classes in Fig. 3? Do the authors test different number of classes? The name of each class is also quite strange. What is the difference between low and lower? Why do not name them clearer way such as low, medium, high, superhigh…? What are the criteria between classes? These issues should be clearly explained.

Response 3: Thanks for your wonderful suggestions. In general, the Natural breakpoint method is a statistical method for grading and classifying according to the numerical statistical distribution law, which can maximize the difference between classes, with Jenks Optimization. There are some natural turning points and characteristic points in any statistical series. These points can be used to divide the research objects into groups with similar properties. Therefore, the split point itself is a good for classification. ArcGIS uses it as the default classification method for this reason. Dividing Figure 2 into 6 categories and Figure 3 into 4 categories is a suitable number of categories obtained when we test in ArcGIS software, which can clearly show the spatial distribution characteristics of the data, which is close to reality, and is easier to analyze and interpret the results. The nomenclature of the categories is named according to the research habits of Chinese academic circles, low value<sub-low value<lower value<higher value<sub-high value<high value.

Point 4: It is still very hard to understand Table 2 and 3. It will be good to make graphs to show the differences between areas and years instead of Table 2 and 3. For the name of region it will be good to indicate which class (classes in fig. 2 and 3) they belong to. It is recommended to categorize or order them according to the classes.

Response 4: Thanks for your great suggestions. We replaced the original Tables 2 and 3 with figures. The names of districts and counties are arranged in the order of names announced in the official statistical yearbook, without human intervention, see Figures 4 and 6.